

# The impact of storage conditions on human stool 16S rRNA microbiome composition and diversity

Lauren V. Carruthers[1,2], Arinaitwe Moses[3], Moses Adriko[3], Christina L. Faust[1,2], Edridah M. Tukahebwa[3], Lindsay J. Hall[4,*], Lisa C. Ranford-Cartwright[1,*] and Poppy H.L. Lamberton[1,2,*]

[1] Institute of Biodiversity Animal Health and Comparative Medicine, University of Glasgow, Glasgow, UK
[2] Wellcome Centre for Integrative Parasitology, University of Glasgow, Glasgow, UK
[3] Vector Control Divison, Ugandan Ministry of Health, Kampala, Uganda
[4] Gut Microbes & Health, Quadram Institute Bioscience, Norwich, UK
* These authors contributed equally to this work.

Corresponding author
Lauren V. Carruthers,
l.carruthers.1@research.gla.ac.uk

## ABSTRACT

**Background:** Multiple factors can influence stool sample integrity upon sample collection. Preservation of faecal samples for microbiome studies is therefore an important step, particularly in tropical regions where resources are limited and high temperatures may significantly influence microbiota profiles. Freezing is the accepted standard to preserve faecal samples however, cold chain methods are often unfeasible in fieldwork scenarios particularly in low and middle-income countries and alternatives are required. This study therefore aimed to address the impact of different preservative methods, time-to-freezing at ambient tropical temperatures, and stool heterogeneity on stool microbiome diversity and composition under real-life physical environments found in resource-limited fieldwork conditions.

**Methods:** Inner and outer stool samples collected from one specimen obtained from three children were stored using different storage preservation methods (raw, ethanol and RNAlater) in a Ugandan field setting. Mixed stool was also stored using these techniques and frozen at different time-to-freezing intervals post-collection from 0–32 h. Metataxonomic profiling was used to profile samples, targeting the V1–V2 regions of 16S rRNA with samples run on a MiSeq platform. Reads were trimmed, combined and aligned to the Greengenes database. Microbial diversity and composition data were generated and analysed using Quantitative Insights Into Microbial Ecology and R software.

**Results:** Child donor was the greatest predictor of microbiome variation between the stool samples, with all samples remaining identifiable to their child of origin despite the stool being stored under a variety of conditions. However, significant differences were observed in composition and diversity between preservation techniques, but intra-preservation technique variation was minimal for all preservation methods, and across the time-to-freezing range (0–32 h) used. Stool heterogeneity yielded no apparent microbiome differences.

**Conclusions:** Stool collected in a fieldwork setting for comparative microbiome analyses should ideally be stored as consistently as possible using the same preservation method throughout.

# INTRODUCTION

Profiling faecal microbiota is now routinely applied to explore relationships between microbiota and host health status (*Young, 2017*). Since stool, including the microbiota, is subject to change post-collection, it is essential that samples are preserved in a way that minimises microbial growth, degradation and contamination to ensure microbial associations being detected in comparative studies are not influenced by storage. The 'gold standard' for storing stool for microbiome analysis is cryopreserving at −80 °C without a buffer (*Vandeputte et al., 2017*). Preservation at −20 °C has also been proposed as appropriate (*Song et al., 2016*), although this may not be ideal for longer term storage (*Bahl, Bergström & Licht, 2012*; *Gorzelak et al., 2015*). Whilst suitable for human studies in high income countries, cryopreservation is often not feasible for large scale projects in remote fieldwork settings, especially in low and middle-income countries (LMIC). Focusing on conditions more likely to be accessible in these settings, several studies have assessed the impact of storage under standard cold chain, that is, +4 °C (*Choo, Leong & Rogers, 2015*; *Lauber et al., 2010*; *Penington et al., 2018*; *Tedjo et al., 2015*), and 'room' (i.e. 25 °C) temperatures (*Cardona et al., 2012*; *Guo et al., 2016*; *Lauber et al., 2010*; *Tal et al., 2017*; *Tedjo et al., 2015*) prior to freezing. These approaches appear to be sufficient to maintain a representative metataxonomic 16S rRNA microbiota community profile in the short-term (up to 14 days post collection). However, to the best of our knowledge, there have been no studies determining the effect of real time temperature fluctuations commonly seen in tropical fieldwork environments. Since geographically separated populations have distinct microbiota compositions (*Lee et al., 2014*; *Yatsunenko et al., 2012*), it is reasonable to hypothesise that the microbiota in stool samples from different communities could also have different rates of abiotic change. Exploring the impact of time-to-freezing on gut microbiota profiles is therefore an important consideration for field studies in tropical LMIC, where the gut microbiome composition is less well established, temperature variation is more difficult to control, and collection standards are difficult to optimise.

Furthermore, access to laboratory consumables and resources are often limited, unreliable and potentially challenging to replenish in remote LMIC locations. Informed and realistic considerations must be made about the best practices for storage of stool samples in such situations to maintain sample integrity. More recently, preservation solutions have been used in an attempt to preserve DNA, and minimise microbial changes in stool after collection. Minimal differences have been reported between different room-temperature storage preservation solutions compared to immediately frozen raw stools (*Blekhman et al., 2016*; *Dominianni et al., 2014*; *Wang et al., 2018*). Another study reported that stool microbiome 16S rRNA profiles stored in preservatives at ambient temperature for three days prior to freezing at −80 °C were significantly different in composition and diversity compared to immediately frozen samples without preservative

(*Choo, Leong & Rogers, 2015*). Although storage preservation was being compared, it is possible that both time-to-freezing and abiotic factors influenced results. Understanding the performance of preservation methods, as well as their impact in combination with time-to-freezing, may be useful in settings susceptible to large temperature fluctuations, where cold storage may be unreliable or unavailable.

Sample heterogeneity is another important consideration when trying to obtain representative microbiota profiles, as previous studies have indicated microbial profiles differ in different parts of the stool sample (*Gorzelak et al., 2015*; *Wesolowska-Andersen et al., 2014*). Therefore, ensuring samples collected are representative and consistent, particularly in fieldwork situations where homogenisation of the stool sample may be difficult due to limited resources, is another sampling consideration.

To address these crucial issues, we explored the influence of time-to-freezing, storage preservation methodology, and stool heterogeneity on microbiome profiles for stool specimens collected from three children within a Ugandan community representative of an LMIC fieldwork setting. Stool donor was found to be the greatest source of microbiota variation. Differences between the preservation method were also observed, but to a lesser extent.

## MATERIALS AND METHODS

### Ethics statement

This study was approved by the University of Glasgow College of Medical Veterinary and Life Sciences Ethics Committee (project code 200160068), the Vector Control Division, Ministry of Health Uganda, Research Ethics Committee (reference: VCDREC/062) and the Uganda National Council for Science and Technology (UNCST-HS 2193). Informed signed or fingerprinted parental or guardian consent, and signed or fingerprinted assent from the study children was obtained prior to participation.

### Sample collection

One stool specimen was collected from three children, aged 12–14, selected at random from Bugoto Lake View Primary School, Mayuge District, Uganda in March 2017. The sample from Child A was collected on day 1, and those from Child B and Child C on day 2. Outer surface, central inner and mixed stool samples (~300 mg each) were taken from each specimen and stored separately in cryovials as raw stool (considered the standard), dispersed in absolute ethanol (approx. stool:ethanol ratio = 1:6) and dispersed in RNAlater (approx. stool:RNAlater ratio = 1:6), then frozen immediately on dry ice. Additionally coarsely homogenised stool from each donor were frozen on dry ice at 1, 2, 4, 8, 16 and 32 h post collection for each storage preservation method (Table S1). Time zero was taken as the time at which all the stool samples, taken from an individual stool specimen, had been processed into all the collection tubes for the relevant conditions to be tested, which was approximately 30 min after defecation. Prior to freezing on dry ice, stool was kept in cooler, shaded, well ventilated, indoor spaces as much as realistically possible. Within 48 h of freezing on dry ice, samples were transferred into a −20 °C freezer and later transported to the University of Glasgow on dry ice for further processing and

analysis. Samples underwent one freeze-thaw cycle (<30 min) during weighing and, to the best of our knowledge, they remained frozen at −20 °C from collection until DNA was extracted approximately 6 months later. Cryovials used in the field containing only ethanol or RNAlater, without stool, were used as negative controls. Samples from two of the children were also stored using OMNIgene.GUT kits (DNA Genotek (*Doukhanine et al., 2014*)) as per manufacturer's instructions, and remained at ambient temperature until DNA was extracted approximately six (four in Uganda and two in the UK) months later (~three times the recommended 60 day stability recommendation for the kit; http://www.dnagenotek.com/us/products/collection-microbiome/omnigene-gut/OMR-200.html). Details of samples and sample codes are shown in Table S1.

## Extraction of DNA from stool

The MPbio FastDNA™ SPIN Kit for Soil (MP Biomedical, Irvine, CA, USA), was used to extract nucleic acids from ~200 mg of stool with minor modifications to the method described by *Alcon-Giner et al. (2017)*, as follows. An attempt was made to exclude large pieces of undigested vegetable matter from stool during the weighing process. Samples were homogenised using a TissueLyser II (Qiagen, Hilden, Germany) at a speed setting of 25, and a 2 min centrifugation was used after addition of binding matrix. DNA concentration was quantified using a NanoDrop 1000 fluorimeter (Thermo Fisher Scientific, Waltham, MA, USA).

## 16S library preparation

A modification of the Illumina 16S metagenomic sequencing library preparation protocol (https://support.illumina.com/documents/documentation/chemistry_documentation/16s/16s-metagenomic-library-prep-guide-15044223-b.pdf) was used to prepare the DNA library. PCR was used to amplify the V1–V2 regions of the 16S rRNA gene, chosen because they were better at detecting bacterial species of interest from stool for future studies (eg. *Bifidobacterium* (*Alcon-Giner et al., 2017*)). The primers used were: 16SV1 forward primer (5′-TCGTCGGCAGCGTCAGATGTGTATAAGAGACAGAGMGTTYGATYMTGGCTCAG-3′) and 16SV2 reverse primer (5′-GTCTCGTGGGCTCGGAGATGTGTATAAGAGACAGCTGCCTCCCGTAGGAGT-3′).

Each reaction was performed in a final volume of 25 μL consisting of 1× KAPA HiFi HotStart ReadyMix (KAPA Biosystems, Wilmington, MA, USA), 0.5 μM of each primer and 12.5 ng of sample DNA. Thermocycler conditions were used as follows: 95 °C for 5 min, followed by 26 cycles of 95 °C for 30 s and 60 °C for 1 min. Samples were then held at 10 °C in the PCR machine, before being stored at 4 °C. $H_2O$ sample controls were included as negative controls during the first round of PCR to monitor non-specific amplification.

Each PCR product was purified by mixing with a $0.90 \times$ PCR product volume of High Prep PCR beads (MAGBIO, Gaithersburg, MD, USA). After a 10 min incubation, sample tubes were placed on a magnetic stand and left until the supernatant became clear. The supernatant was then removed and the beads were washed twice with freshly prepared 80% ethanol, and then left to dry for 15 min to allow residual ethanol to evaporate. The sample tubes were removed from the stand and the beads were

then resuspended in 20 μL Tris buffer pH 8.5, and incubated for 2 min before being placed back on the magnetic stand. Once clear, the supernatant was transferred to a fresh tube and the DNA concentration quantified using the Quant-iT PicoGreen dsDNA Assay (https://assets.thermofisher.com/TFS-Assets/LSG/manuals/mp07581.pdf) (Thermo Fisher Scientific, Waltham, MA, USA).

A second PCR step was then used to barcode each sample. PCR reactions were performed in a final volume of 50 μL consisting of 1× KAPA HiFi HotStart ReadyMix, 5 μL of each of two Nextera XT Index Kit Set A (Illumina, San Diego, CA, USA) indices, with each sample having a unique combination, and 10 ng of post-PCR1 sample DNA. Thermocycler conditions used were as follows: 95 °C for 3 min; followed by eight cycles of 95, 55 and 72 °C for 30 s each; with a final step of 72 °C for 5 min. Samples were then held at 10 °C in the PCR machine, before being stored at 4 °C.

Samples were cleaned with High Prep PCR beads as described above and then combined to form an equimolar sample library. The Wizard SV Gel and PCR Clean-Up System Kit (Promega, Madison, WI, USA) was used to purify the DNA library prior to sequencing, as per manufacturer's instructions (https://www.promega.co.uk/-/media/files/resources/protocols/technical-bulletins/101/wizard-sv-gel-and-pcr-clean-up-system-protocol.pdf?la=en) using a band size of ~435 bp. DNA concentration was then measured using a Bioanalyser 2100 (Agilent, Santa Clara, CA, USA).

## Sample sequencing and analysis

Samples were sequenced using the Illumina MiSeq platform (Glasgow Polyomics, Glasgow, UK) with 2 × 300 bp paired-end read lengths with up to 100,000 reads per sample (MiSeq V3 600 cycle kit; Illumina, San Diego, CA, USA). Using cutadapt software (*Martin, 2011*) in Python version 2.7, barcode sequences were removed, reads trimmed to a minimum quality score of 20, and then reads less than 250 bp in length were discarded (Code S1). Forward and reverse reads were combined using PANDAseq (*Masella et al., 2012*) for each sample before all files were merged into one file containing all samples (Code S1). Quantitative Insights Into Microbial Ecology (QIIME) software version 1.9.1 (*Caporaso et al., 2010*) in Python version 2.7 was used to analyse the data. Operational Taxonomic Units were assigned with 97% clustering to the Greengenes database version 13.8 (*DeSantis et al., 2006*) for 16S rRNA gene alignment. Sequences aligning to mitochondria or chloroplast sequences were screened for and removed from the dataset. Custom scripts in QIIME were used to analyse relative taxonomic abundance, and alpha and beta diversity measures (Code S1) at a sequencing depth of 10,000 reads per sample. Pairwise comparisons of beta diversity measures (weighted (*Lozupone et al., 2007*) and unweighted (*Lozupone & Knight, 2005*) UniFrac) were made using 999 Monte Carlo permutations (MCP). The linear discriminant effect size (LEfSe) (*Segata et al., 2011*) algorithm was performed to identify taxonomic groups associated with the variables measured ($p < 0.01$, linear discriminant analysis (LDA) score ($\log10 \geq 2$)). To be included in the results, each variable must have met the inclusion criteria ($p < 0.01$, LDA score ($\log10 \geq 2$)) within each child, as well as when averaged across all three children. Higher taxonomic levels were excluded where it was assumed that a lower taxonomic level

was accountable for the observed change. These situations were where a higher taxonomic level had a less significant or equal change in relative abundance compared to a lower taxonomic level classified to the higher taxonomy by LEfSe analysis. However, if the higher taxonomic level had a more significant $p$-value it was retained. Kruskal–Wallis tests to compare read counts (significantly different if $p < 0.05$) were performed in R version 3.4.2 (*R Core Team, 2017*) and graphs were generated using the ggplot2 package (*Wickham et al., 2018*). All data are provided as Supplemental Information.

Alpha diversity scores (species richness, Shannon and Simpson), generated using standard parameters in QIIME 1.9.1, were analysed by generating linear mixed effect models using the lme4 package (*Bates et al., 2018*) in R version 3.4.2 (*R Core Team, 2017*) to identify important predictors of alpha diversity. The lmerTest package (*Kuznetsova, Brockho & Christensen, 2017*) was used to determine the significance of these model components. Two maximal models were constructed and included all the fixed effects (preservation method, time-to-freezing and stool region) and their interactions with child replicate as a random effect, the first included time as a continuous variable and the second included time as a factor. Backward elimination was used for sequential removal of non-significant variables, to obtain the minimal statistically significant model (*Burnham, Anderson & Huyvaert, 2011*) (Code S2).

Taxonomic abundance graphs and LEfSe plots generated with QIIME were recreated in R using the ggplot2 package (*Wickham et al., 2018*). Principal coordinates analysis (PCoA) plots were generated using Emperor Software (*Vázquez-Baeza et al., 2013*) within QIIME.

Due to the low number ($n = 2$) of OMNIgene.GUT samples taken, and because OMNIgene.GUT samples were only taken from two out of the three children, these samples were excluded from the above analyses and analysed separately.

## RESULTS

### Samples processing and microbiome sequencing

In total 87 stool samples were collected for analysis and libraries prepared. After sample exclusion, trimming and alignment (Figs. S1 and S2; Table S1) there was an average of 67,575 (range 19,083–466,807) reads per sample ($n = 85$).

### Microbiome profiles vary between individual children

Each child had a distinct microbiome signature (Figs. 1A and 1B) that was apparent at all taxonomic levels, from phylum (Fig. 2A; Table S2) to genus (Fig. 2B; Data S1) level regardless of the preservation method used and the time-to-freezing duration. The most abundant phyla were Bacteroidetes in child A (40.7%) and child B (36.9%), and Firmicutes in child C (34.1%); followed by Firmicutes in child A (40.1%), Proteobacteria in child B (30.2%) and Bacteroidetes in child C (28.7%). LEfSe identified several bacterial taxa significantly associated with each individual child (Data S2). PCoA analysis using qualitative (presence/absence) differences (unweighted UniFrac, Fig. 1A) confirmed that the clustering of bacterial sequences within individual children was significantly different (MCP for all child comparisons $p \leq 0.001$) (Fig. 1A). Children were also

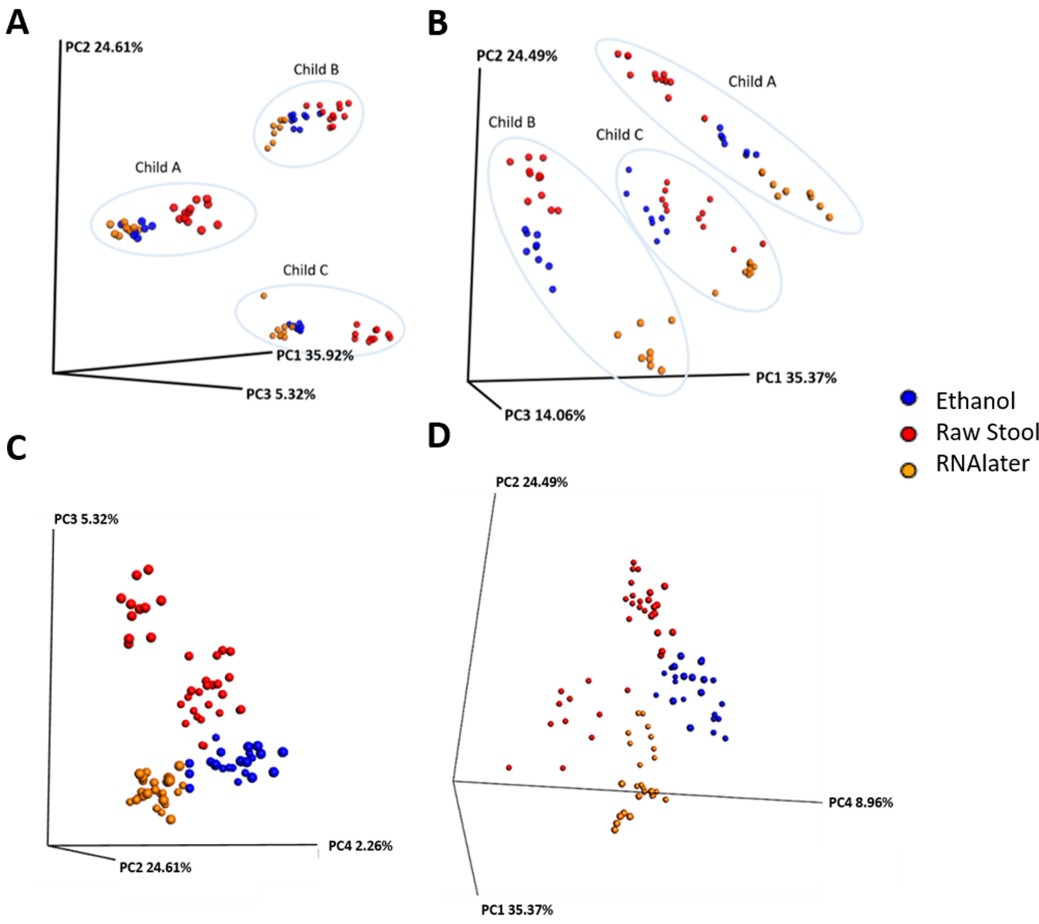

**Figure 1 Samples cluster by individual and storage method using principal coordinate (PC) analysis of unweighted (A) and (C) and weighted (B) and (D) unifrac measures.** Ellipses enclose samples from the same individual (A) and (B). Storage method: Red = raw stool, Blue = ethanol and Orange = RNA-later. PC1, PC2 and PC3 (A) and (B); PC2, PC3 and PC4 (C); and PC1, PC2 and PC4 (D).

significantly different by relative abundance weighted UniFrac (MCP for all child comparisons $p \leq 0.001$) (Fig. 1B).

## Microbiome profiles vary by stool storage method used

The samples stored as raw stool had a mean average of 80,822 reads per sample (range 26,270–466,807; 35 samples), samples in ethanol had a mean average of 62,983 reads per sample (range 24,215–140,356; 24 samples), and samples in RNAlater had a mean average of 53,981 reads per sample (range 19,083–100,934; 26 samples) (Fig. S3). The number of read counts was not significantly different between preservation methods using a Kruskal–Wallis test.

The within-individual variation between samples stored under different preservation methods was less than that observed between individuals (Figs. 1A, 1B and 2). Intra-storage preservation method microbiota abundance compositions were similar over the time points examined (0–32 h) for each of the preservation methods used (Fig. 2),

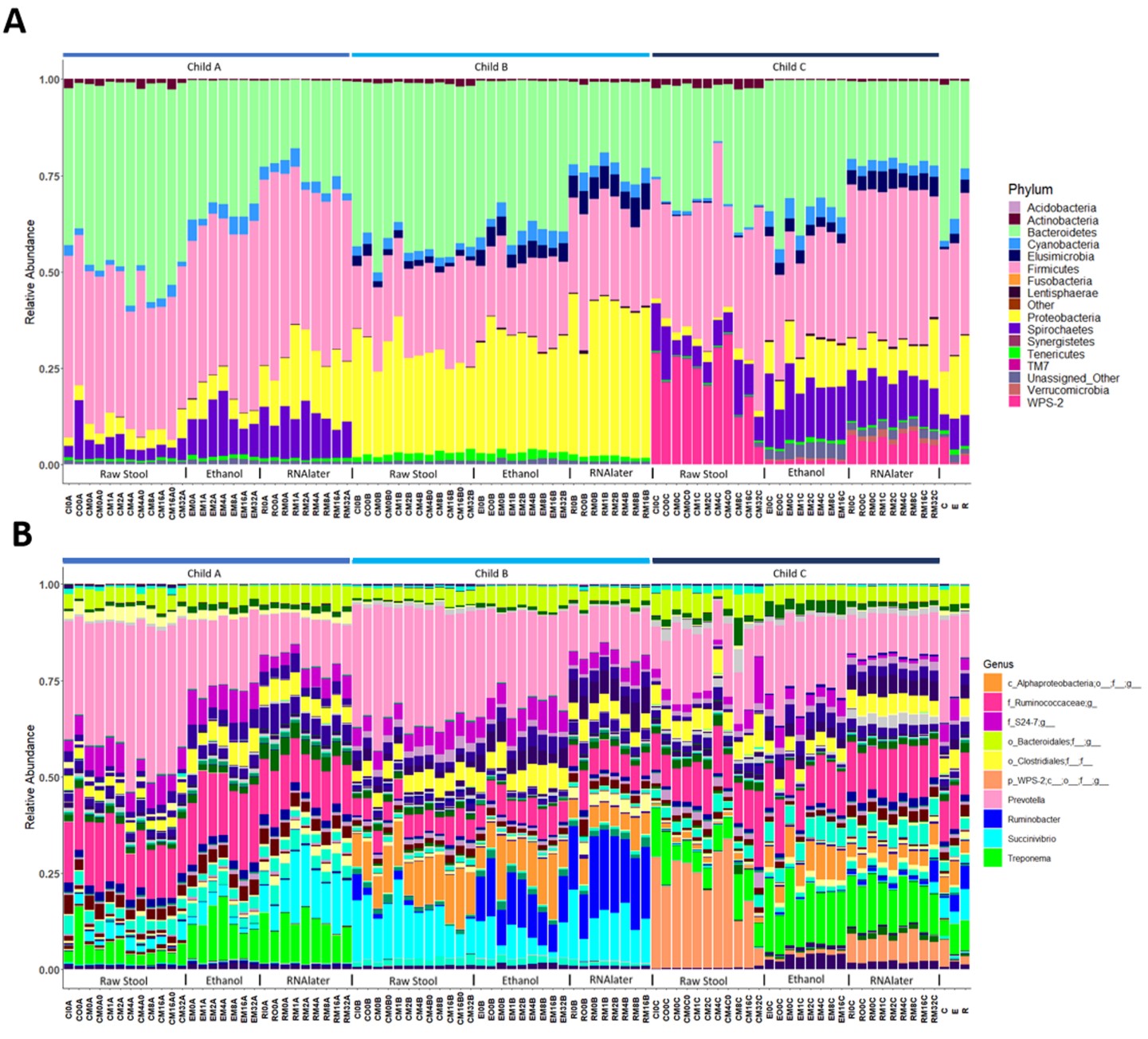

**Figure 2** **Relative bacterial abundance patterns of samples at the phylum (A) and genus (B) level varies between children and storage technique used within each child.** The top 10 genera are included in the legend (B); where a genus name was not provided the lowest taxonomic resolution has been used where p, phylum; c, class; o, order; f, family. For a full annotation of the genus legend refer to the *Supplemental Genus Legend*. For a full description of sample codes refer to Table S1. Individual letter descriptors indicate the mean average relative abundances of raw stool (C), ethanol (E) and RNAlater (R) storage preservation across all children, time points and stool regions.

suggesting relative stability. There were twelve taxonomic groups significantly associated with raw stool (all $p < 0.01$, LDA score (log10 ≥ 2), Data S3). Eight and eleven taxonomic groups were positively associated with ethanol and RNAlater storage respectively, compared to raw stool alone (all $p < 0.01$, LDA score (log10 ≥ 2) (Fig. 3; Data S3). Seven of

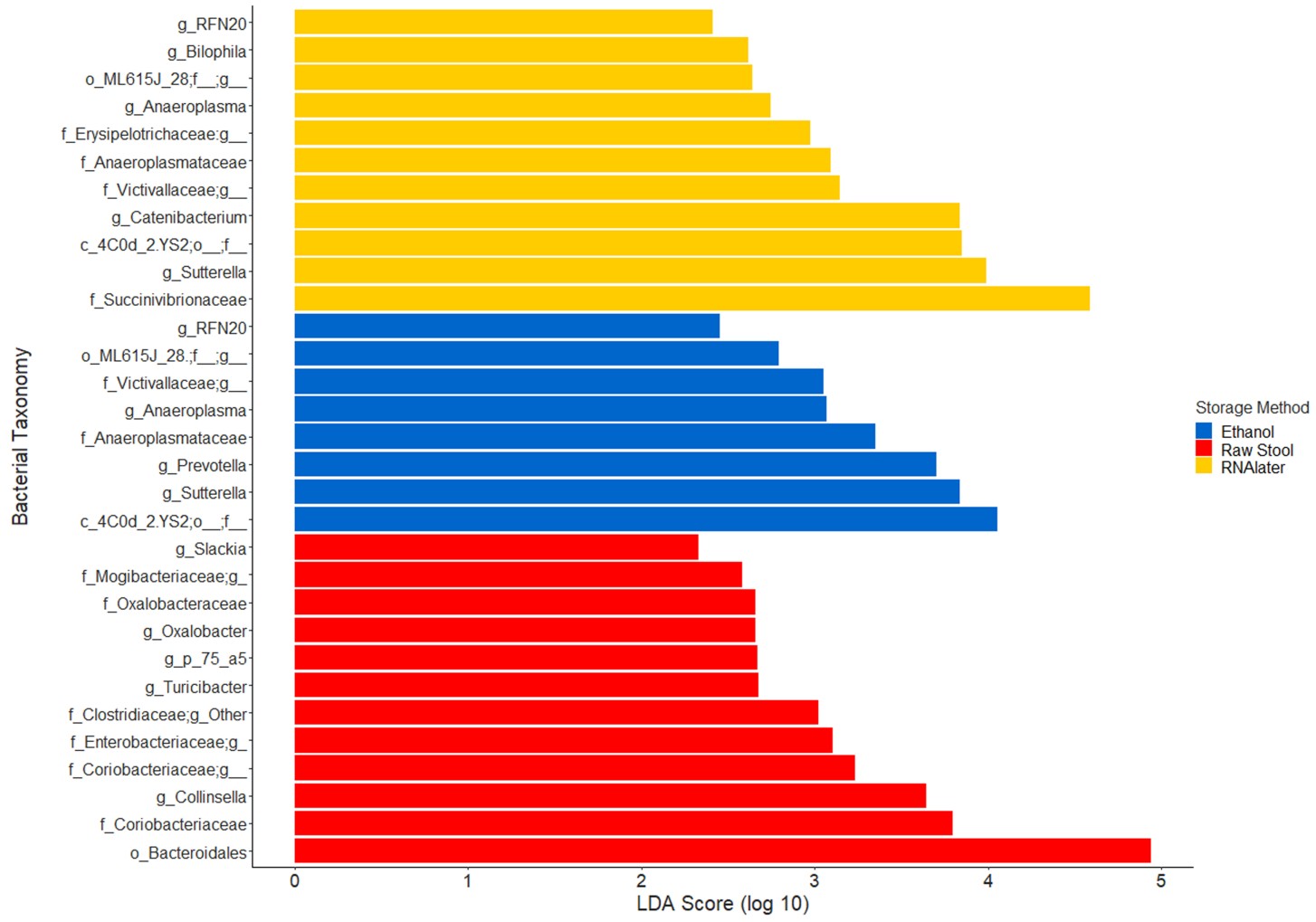

**Figure 3 Bacterial groups significantly positively associated with different preservation methods of storage by linear discriminant analysis effect size (LEfSe).** Raw Stool, raw stool vs. ethanol and RNAlater; Ethanol, ethanol vs. raw stool only; and RNAlater, RNAlater vs. raw stool only. The significance parameters (LDA Score (log10 ≥ 2), $p < 0.01$) were met within each individual and when averaged across all three children to be included. The most descriptive taxonomic resolution is provided unless a higher taxonomy was more significant, in which case both are shown (For all information refer to Data S3).               

the taxonomic groups that were significantly elevated in relative abundance were shared in the samples stored in ethanol and RNAlater (Fig. 3; Data S3).

Microbiome diversity under different preservation methods were found to differ significantly within each child by qualitative unweighted UniFrac analysis (MCP; all comparisons $p \leq 0.001$). The unweighted Unifrac distances within ethanol samples were found to be significantly different compared to within RNAlater samples (child A: $p = 0.003$, child B: $p = 0.041$, child C: $p = 0.035$) or within raw stool samples (child A: $p = 0.02$, child B: $p = 0.038$, child C: $p \leq 0.001$). UniFrac metrics within raw stool samples were not significantly different to metrics within RNAlater samples apart from in child C ($p \leq 0.001$). MCP analysis of unweighted UniFrac comparisons across all three children also revealed significant differences between raw stool and RNAlater storage methods (raw stool vs. raw stool:raw stool vs. RNAlater, $p = 0.005$; RNAlater vs. RNAlater:RNAlater

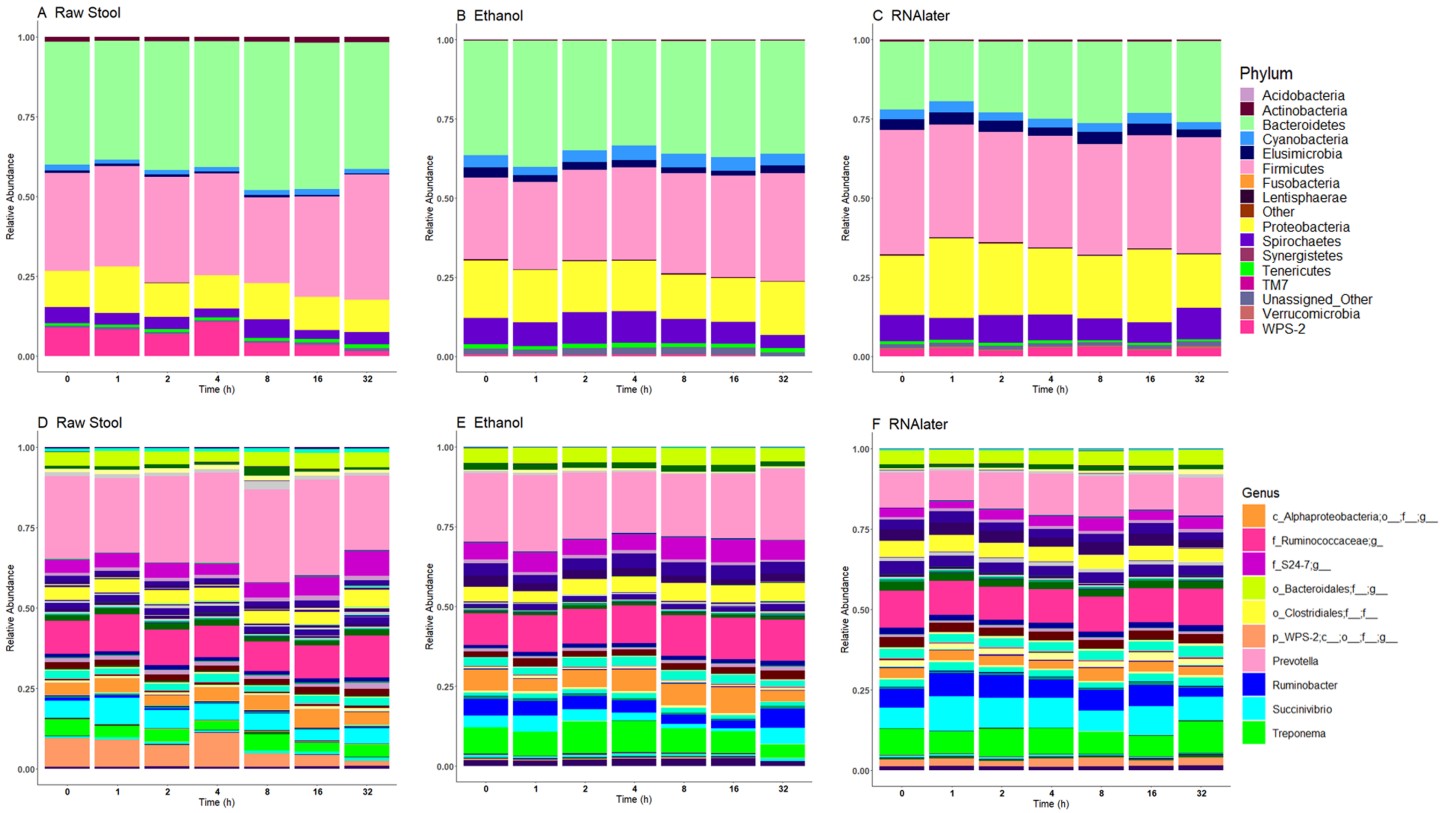

**Figure 4** **Microbiome relative abundance profiles remain relatively stable over time across all storage methods used at the phylum (A–C) and genus (D–F) levels.** The top 10 genera are included in the legend (B); where a genus name was not provided the lowest taxonomic resolution has been used where p, phylum; c, class; o, order; f, family. For a full annotation of the genus legend refer to the *Supplemental Genus Legend*. Raw stool (A) and (D), 100% ethanol (B) and (E) and RNAlater (C) and (F). Samples were averaged across all three children and include all stool regions.

vs. raw stool, $p = 0.01$), and raw stool and ethanol storage (raw stool vs. raw stool: raw stool vs. ethanol, $p = 0.026$) (Fig. 1C) despite distinct separation by child (Fig. 1A).

When UniFrac measures were weighted by relative sequence abundance within each child and as an average of all three children, all preservation method comparisons by MCP were found to be significantly distant from each other (for all comparisons $p \leq 0.001$) (Figs. 1B and 1D). Separation within raw stool samples was also found to be significantly different compared to separation within ethanol storage (MCP; child A: $p = 0.017$, child B: $p = 0.041$, child C: $p \leq 0.001$, all children: $p = 0.005$). In child C within RNAlater metrics were found to be significantly different by MCP compared to within raw stool metrics ($p \leq 0.001$) however, this was not observed in child A or child B. No significant differences were observed by MCP between within RNAlater and within ethanol weighted UniFrac metrics.

## Microbiome profiles remain relatively stable over time

Relative bacterial abundance and composition remained relatively stable over time-to-freezing across all storage techniques by LEfSe analysis when time-to-freezing was included as a continuous variable (Fig. 4). However, LEfSe analysis indicated a significantly

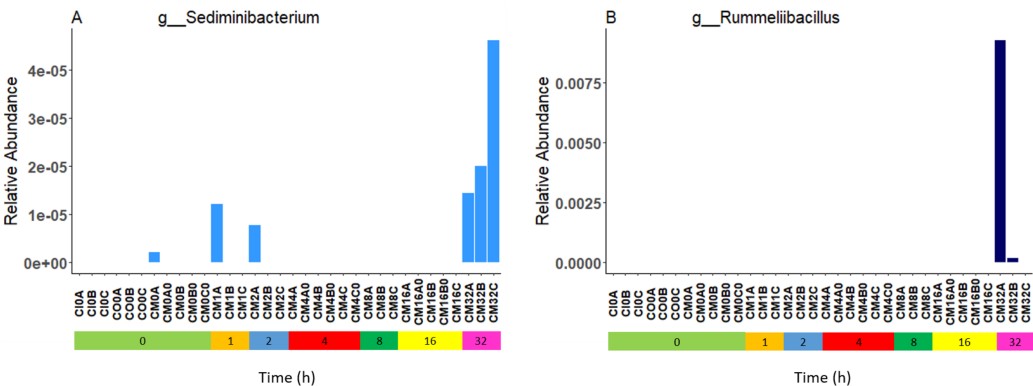

**Figure 5 Bacterial groups identified to be significantly more abundant in raw stool samples at 32 h time-to-freezing by LEfSe analysis:** *Sediminibacterium* **(A) and** *Rummeliibacillus* **(B).** For samples to be included they must meet the following criteria: LDA Score (log10 ≥ 2), $p < 0.01$.

increased relative abundance of two bacterial genera in raw stool samples at 32 h time-to-freezing when time-to-freezing was included as a categorical variable: *Sediminibacterium* ($p = 0.0016$, LDA (log10) = 2.35) (Fig. 5A) and *Rummeliibacillus* ($p = 0.0012$, LDA (log10) = 3.25) (Fig. 5B). No significant categorical time-to-freezing effects were identified in ethanol or RNAlater samples by LEfSe analysis. No apparent time clustering was observed by PCoA using UniFrac metrics. No significant differences by weighted or unweighted UniFrac metrics using MCP were observed when comparing 0 h sample metrics to any other time-to-freezing time point or vice versa.

## No significant differences in microbiome composition were observed between inner, outer and mixed regions of stool samples

Microbiome profiles of the mixed stool samples were similar to inner and outer stool samples, from phylum through to genus level (Fig. 6). Significant differences between stool regions were not observed in this study by LEfSe regardless of whether the data were analysed by child, storage preservation method or as a whole. No significant associations were generated when LEfSe analysis was performed and no apparent clustering was identified by PCoA analysis of UniFrac diversity metrics. Weighted and unweighted UniFrac comparisons showed no significant differences using MCP.

## Modelling indicates storage method influences stool alpha diversity

Linear mixed effect models were constructed to detect variables associated with alpha diversity metrics (Code S2; Table S3). None of the final models identified stool region or time (included as a fixed or a continuous variable) to be significant predictors of alpha diversity. Individual children included as a random effect alone was the only variable shown to influence Shannon diversity and Simpson diversity.

Shannon ~ 1 + (1|child)
Simpson ~ 1 + (1|child)

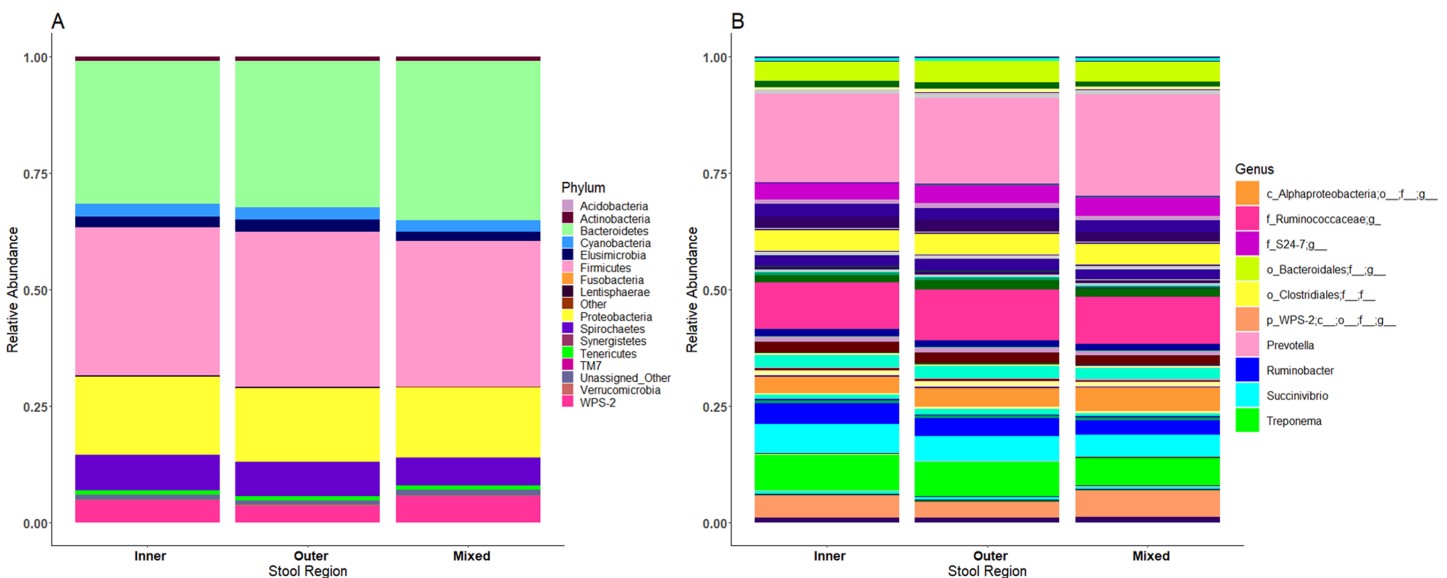

**Figure 6 Microbiome profiles remained stable across stool regions.** Phyla (A) and genera (B). The top 10 genera are included in the legend; where a genus name was not provided the lowest taxonomic resolution has been used where p, phylum; c, class; o, order; f, family. For a full annotation of the genus legend refer to the *Supplemental Genus Legend*.

Preservation method was identified to be a significantly important model component for the model predicting species richness ($p$ = 2.871e−13) with stool stored in RNAlater having the highest average richness, followed by raw stool and stool stored in ethanol (Table S3). Compared to the null species richness model, 50.2% of species richness variation was accounted for by the final species richness model.

Species richness ~ preservation method + (1|child)

## OMNIgene.GUT sample performance

Samples stored using OMNIgene.GUT kits had a mean average of 77,089 reads per sample (child B = 102,227; child C = 51,951). The samples clustered by PCoA analysis to the relevant children from which they were taken (Fig. S2). Relative abundance profiles were also representative of the microbiome profiles from each child (Fig. S4).

## DISCUSSION

As has been previously observed in stool microbiome studies (*Blekhman et al., 2016; Carroll et al., 2012; Dominianni et al., 2014; Guo et al., 2016; Lauber et al., 2010; Penington et al., 2018; Wang et al., 2018*), the sample donor was found to be the greatest predictor of gut microbiome variation amongst the variables studied here. Samples were identifiable to each child regardless of storage method, time-to-freezing or stool region (Figs. 1A, 1B and 2) (*Blekhman et al., 2016; Carroll et al., 2012; Dominianni et al., 2014; Guo et al., 2016; Lauber et al., 2010; Penington et al., 2018; Wang et al., 2018*). Individual child (included as a random effect) was also the only predictor of Shannon diversity and Simpson diversity from the variables measured. Multiple factors, including diet (*David et al., 2014*) and demographics (*Yatsunenko et al., 2012*), not recorded in this study, have

been shown to influence microbial status within an individual, and each individual will have a unique combination of contributing factors. Individuality is therefore an important consideration when planning comparative microbial studies (i.e. between healthy and diseased states) to ensure enough participants are recruited into studies so that the obtained data are informative about the question of interest.

Although the microbiome diversity under different preservation methods clustered by child, both weighted and unweighted UniFrac metrics also indicated that samples stored by different preservation methods were significantly distant within each of the children (Figs. 1A and 1B). PCoA of UniFrac metrics further revealed clustering by preservation method despite child variation (Figs. 1C and 1D), suggesting each preservation method acts similarly across each child. Despite significant differences between within-preservation-method UniFrac metrics, there was no trend in preservation method performance across each of the children. This could be due to unique microbial profiles of the children being suited to different types of storage or attributed to the randomness between samples taken within a specimen, which may also account for the significant differences in UniFrac metrics observed in child C. Modelling also identified that preservation method had a significant effect on species richness. Significant differences in the microbial profiles from stool stored in RNAlater and ethanol were identified when compared to samples stored raw, that were considered 'gold standard' for this study (Fig. 3). These differences appeared to remain relatively stable across time-to-freezing (Fig. 4) and were evident even at time zero, when samples were first frozen by 30 min post-defecation, suggesting that changes occurred rapidly, within a few minutes, after the addition of stool to preservative. All of the bacterial levels correlated to ethanol and/or RNAlater preservation identified by LEfSe, of which the two methods shared seven groups of the eight and 11 groups respectively (Fig. 3), were associated with some form of anaerobic metabolism. Anaerobic bacteria are possibly over-represented or better preserved than aerobic species in stool stored using these methods. Preservative exposure therefore may influence microbial profiles obtained from stool via a common physical mechanism, which favours the preservation of some bacterial taxa over others, making stool stored in various preservatives more similar in microbial structure and comparable to each other. This is in agreement with a study that found samples stored in preservatives were more likely to cluster together by PCoA, based on Bray-Curtis similarity distances (Choo, Leong & Rogers, 2015). Alternatively, it is possible that PCR product amplification of certain species was altered by residual ethanol or RNAlater salts despite care being taken to limit these PCR contaminants during the DNA extraction process. RNAlater has been reported to reduce DNA yield by qPCR (Gorzelak et al., 2015) and 16S rRNA DNA amplification purity (Dominianni et al., 2014) in microbiota studies. However, microbiota variation between samples within preservation method groups was low (Figs. 1, 2 and 4) and DNA concentrations were standardised across the samples in our study, suggesting that at least some of the associations observed are due to the stool preservation method.

Irrespective of preservation techniques, microbiome profiles remained adequately stable for up to 32 h in tropical ambient temperatures when compared to their baseline

(0 h, frozen by 30 min post-defecation), with only two minor, albeit significant, changes in relative abundance arising by 32 h in raw stool samples when time-to-freezing was included as a categorical variable (Fig. 5). The identified increase in one of these, *Rummeliibacillus* however, is likely strongly influenced by one sample, CM32A (Fig. 5B; Table S1), making further studies necessary to determine the reproducibility and impact of this finding. Sampling at more regular intervals between 16 and 32 h would reveal if these increases are continuous over time and when they might start to occur.

Models did not identify time to be a significant predictor as a continuous or factorial component for species richness, Shannon diversity or Simpson diversity. These findings are in agreement with *Tedjo et al. (2015)* and *Tal et al. (2017)* who found no significant differences in diversity measure scores after 24 h (Shannon and Chao1) and 96 h (Shannon, Simpson and Chao1) of room temperature storage respectively. Storage at room temperature did significantly reduce weighted Shannon and Weaver diversity scores by 17% after 8 h at room temperature in another study (*Ott et al., 2004*). Diversity scores however, should always be considered in the context of specific bacteria profiles since the presence and absence of bacteria could change over time but the derived diversity score could remain stable.

No significant observations in microbial profiles were identified between the stool regions, a finding in contrast to previous studies reporting differences using qPCR (*Gorzelak et al., 2015*) and associations between microbial richness and stool consistency (*Vandeputte et al., 2016*). The Bristol stool chart (*O'Donnell, Virjee & Heaton, 1990*) defines seven levels of stool consistency and water content from type 1 (solid lumps) to a type 7 (watery liquid). It is plausible that inner and outer stool regions at the higher end of the scale are likely to be more uniform than stool at the lower end of the scale, and more difficult to define inner and outer stool regions at higher stool values. Stool samples collected from the Ugandan children for this study, although not formally graded, commonly fell into the higher end of the Bristol stool chart guide. Classifying stool specimens prior to sectioning may be a useful factor to explore in future work, along with other associations such as diet or health status. Stool size may also impact heterogeneity with the inner and outer regions of 'larger' stools being more distinguishable. This may explain why *Gorzelak et al. (2015)* and *Vandeputte et al. (2016)* obtained significant differences as their samples were collected from adults, who presumably produced larger stool specimens at the lower end of the Bristol stool chart than the LMIC child samples collected in this study. Whilst we did not see any differences associated with stool region, suggesting crude mixing is sufficient to maintain a representative microbiome in situations where specialised equipment is unavailable, the number of specimens collected in this study was small ($n = 3$). Therefore, there may not have been enough replicates to detect changes in stool heterogeneity in this study, and more samples ranging in different sizes may need to be studied to fully understand the impact of stool heterogeneity.

## CONCLUSIONS

Stool samples collected for microbiome analyses are subject to biological change upon exposure to abiotic differences in the environment. This study examined the impact of

different stool storage conditions on the human gut microbiome composition in a tropical LMIC, resource-limited setting. Stool donor accounted for the greatest amount of variation seen in the gut microbiota. Stool storage preservation method significantly influenced the bacterial profiles obtained, however, all samples remained identifiable to their child of origin. Stool stored at ambient temperature for up to 32 h did not significantly influence diversity and had minimal changes upon microbiota composition, which remained relatively stable across time-to-freezing regardless of preservation method used. No apparent differences were observed between outer, inner or mixed stool regions taken however, sample size was small. Overall, comparative studies involving stool storage for microbiome analysis should be performed as consistently as possible in the tropical resource-limited settings, using the same preservation method throughout.

## ACKNOWLEDGEMENTS

We would like to thank our Vector Control Division fieldwork driver Lugigana 'Fiddi' Andrew for his dedication and enthusiasm. Thank you to the community of Bugoto, Uganda, for making us feel welcome, and to the children for participating in our study. Thank you to Dr. David McGuinness at Glasgow Polyomics for providing bioinformatics training and technical support.

The authors Lindsay J. Hall, Lisa C. Ranford-Cartwright and Poppy H.L. Lamberton contributed significantly and equally to the design of the study as well as to the analysis and interpretation of the data and drafting of the manuscript, but in different ways based on their different expertise.

### Funding

This work was supported by Wellcome (105614/Z/14/Z), a Lord Kelvin Adam Smith Fellowship to Poppy H.L. Lamberton and a studentship to Lauren V. Carruthers (IBAHCM_stipend_144536) from the University of Glasgow, and the European Research Council Starting Grant (SCHISTO_PERSIST_680088). The funders had no role in study design, data collection and analysis, decision to publish, or preparation of the manuscript.

### Grant Disclosures

The following grant information was disclosed by the authors:
Wellcome: 105614/Z/14/Z.
Lord Kelvin Adam Smith Fellowship to Poppy H.L. Lamberton and a Studentship to Lauren V. Carruthers: IBAHCM_stipend_144536.
University of Glasgow, and the European Research Council Starting Grant: SCHISTO_PERSIST_680088.

### Competing Interests

The authors declare that they have no competing interests.

## Author Contributions

- Lauren V. Carruthers conceived and designed the experiments, performed the experiments, analysed the data, prepared figures and/or tables, authored or reviewed drafts of the paper, approved the final draft.
- Arinaitwe Moses performed the experiments, authored or reviewed drafts of the paper, approved the final draft.
- Moses Adriko performed the experiments, authored or reviewed drafts of the paper, approved the final draft.
- Christina L. Faust conceived and designed the experiments, authored or reviewed drafts of the paper, approved the final draft.
- Edridah M. Tukahebwa conceived and designed the experiments, authored or reviewed drafts of the paper, approved the final draft.
- Lindsay J. Hall conceived and designed the experiments, analysed the data, authored or reviewed drafts of the paper, approved the final draft.
- Lisa C. Ranford-Cartwright conceived and designed the experiments, analysed the data, authored or reviewed drafts of the paper, approved the final draft.
- Poppy H.L. Lamberton conceived and designed the experiments, authored or reviewed drafts of the paper, approved the final draft.

## Human Ethics

The following information was supplied relating to ethical approvals (i.e., approving body and any reference numbers):

The University of Glasgow College of Medical Veterinary and Life Sciences Ethics Committee (project code: 200160068), the Vector Control Division, Ministry of Health Uganda, Research Ethics Committee (reference: VCDREC/062) and the Uganda National Council for Science and Technology (reference: UNCST-HS 2193) reviewed and approved the methods for this study.

## Data Availability

Raw sequencing data is available in the European Nucleotide Archive, accession number: PRJEB32925.

## Supplemental Information

Supplemental information for this article can be found online at http://dx.doi.org/10.7717/peerj.8133#supplemental-information.

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
