# Peer review of "The impact of storage conditions on human stool 16S rRNA microbiome composition and diversity"

_PeerJ, doi:10.7717/peerj.8133_

## Round 0.1 · original submission · Major Revisions

Dear Lauren and co-authors,

I have now received three detailed reviews of your manuscript. While reviewers recognised the value of this study, they have all listed a number of major issues that need to be carefully revised. In particular, the very low sample set used in the study seems problematic and require running additional statistical analyses and possibly toning down or adapting your conclusions (see reviewers 1 and 2). Reviewer 3 suggested additional proofreading of the manuscript, clarification of the methods section, and improving the Figures.

I will be looking forward to receiving your revised manuscript along with a point-by-point response to the reviewers comments.

With kind regards,
Xavier

Reviewer 1 ·

Basic reporting

Well written and good literature review.

Experimental design

Good technical approach in an important topical area of research

Validity of the findings

The sample size for the study is too small to be meaningful

Additional comments

The authors set to identify the effect of storage conditions on the microbiome of three children from Uganda. The authors used five different collection methods, including raw, ethanol, RNAlater and OMNIgene.GUT. The authors then analysed the 16s rRNA poriles using the V1-V2 regions. There were significant differences were observed in the composition and diversity between preservation methods. However, they could still observe clustering that was driven my child rather than sample preparation method. The authors are particularly interested in how temperature and storage could affect the microbiota and how this relates to samples from field studies, where there are more likely to be fluctuations in the storage collection and conditions which may alter the microbiota. The authors address important questions about the variation in samples depending on collection and storage before processing. The data they present shows the importance of standardizing collection within a study.

Major comments:
The n=3 for most of the samples (raw, ethanol, RNAlater) and n=2 for the OMNIgene.GUT is very low. How robust is this?
Line 231~: “In total 87 stool samples were collected for analysis” >How are these 87 comprised? Are these different times of the 3 samples, in the different storage methods? Please insert a table of these samples.
Figure 1: All children have a distinct microbiome by PCA (Line 236), however on figure 1 you also see a clustering of the RNA later samples. (Fig 1b). Please discuss the clustering – there is similarity between child B and A for ethanol and raw too. (this is partly mentioned in lines 265-267)
Figure 3: the mixture of genera, phylum, family and order is not clear. This needs to be separated out by taxonomic levels
The authors should carry out ANOSIM analysis to determine whether the differences in the population are due to the child or the method.
Lines 257-285: Please quantify this conclusion that inter-storage preservation methods were relatively stable over time. A population analysis over time would answer this.
Lines 277-279: this is most likely not an increase in relative abundance of these genera (Sediminibactreium and Rummeliibacillus), but due to a drop in other taxa which then unmasks this difference. What is the alpha and beta diversity at these time points?
Minor points:
Methods
Line 185 : “and then reads less 186 than 250 bp in length were discarded” What was the length of the V1-V2 PCR? The forward and reverse reads should cover the 453bp fragment?

Line 201-203: “Higher taxonomic groups were 201 excluded where it was assumed that a lower taxonomic group was accountable for the observed 202 change. These situations were where a higher taxonomic group had a less significant or equal 203 change in relative abundance compared to a lower taxonomic group classified to the higher 204 taxonomy by LEfSe analysis.”
This needs further clarification and reasoning.
Line 222: “Principal Coordinates Analysis (PCoA) plots” or principle component analysis?
Lines 311-315: the OMNIgene was only performed on two samples to this is too few to analyse

·

Basic reporting

The manuscript is well written.

Experimental design

Generally speaking, the methods are well described and experiments are well conducted. One limitation is the small sample size. (Only three individuals)

Validity of the findings

I may have missed the Accession numbers, but it does not appear that the data sets from this work have been uploaded to a public database (i.e., 16s sequences FASTQ data).

Additional comments

Generally speaking, the analyses the authors did are pretty straightforward .A moderate degree of revision is needed. Please consider the following comments intended to improve the manuscript.

A more nuanced conclusion seems desirable. A lot of literature already compared 16S fecal sample collection methods/storage conditions. What is the major difference between this study and previous 16S methodology comparison studies? What’s the unique/new finding of this study?
Also, the authors could further discuss the meaning of their findings. Why the tested methods gave different results? What is the mechanism?

Other specific comments:

- The QIIME-OTU picking method (with greengenes 13.8 database) which the authors current using is acceptable, but a little bit old, especially for this kind of methodology paper. Since the QIIME2 platform is public available for a long time (a couple of years), it is recommended to use the better algorithm/pipeline, like deblur- sOTU method.

- According to supplement FigS1, the observed OTU curves did not reach plateau even at 10000 sequences per sample. So it would be good to explain the rarefaction level which used for the downstream analyses , at least in the Method section.

- Most of the recent 16S microbiome studies are targeting on V4 region , or V3-V4 regions. Please explain why your targeting on V1-V2 regions in the discussion section.

- For the alpha- diversity analysis, other indices (eg. PD-whole tree) are needed to better describe more aspects/features of the data. Never the less, these alpha- diversity analyses should be performed at multiple level (eg. genus level, and otu level).

- Fig.1. Title. For beta-diversity weighted/ unweighted UniFrac distance analyses, did the authors use Principal Coordinates Analysis (PcoA) rather than Principal component analysis (PCA)? Please clarify. Here PcoA is recommended.

- Fig.1. For 3D figures, it could be misleading if these figures are not displayed in exact same angle and direction. It is strongly recommended to display Fig1A and Fig1B in exact same manner. Further more, additional analyses like PERMANOVA should be used to provide more evidence.

- Fig. S2. It is strongly recommended to display FigS2A and FigS2B in exact same manner. And add the PERMANOVA R square value to give more information.

- Fig.2B. Should add the figure legend to show the name of these genera, at least the top 10 genera. Put these information in supplemental is not good enough.

- Fig.3. It is not informative to put all different taxa level together and order them by LDA score value. It would be much better to separate this figure by taxa level . An additional phylogenetic tree with highlighted significant taxa would be even better.

- Line23. Typo. “However…”
- Line 115. Please double check. “300g each”??? The reader will be shocked.
- Line 233, “In total 87 stool samples……. there was an average of 233 67,575 (range 19,083 – 466,807) reads per sample (n = 85)”. It looks like two samples were excluded. If so , what's the criteria? Please describe.
- Line306, “Compared to the null species richness model” ,about species richness model, please show more details of the modeling results in Table S6.

Reviewer 3 ·

Basic reporting

The study "The impact of storage conditions on human stool 16S rRNA microbiome composition and diversity" is interesting and very important addition to understand how the method selection affects the results. In general the language is clear, but polishing is needed in some parts. The structure of the article is fine, but some figures need efforts so they would not seem just like a default outputs of a program. Also, supplementary material are very raw!
Raw data not shared! Did not find any notes where is the sequencing data deposited.

Experimental design

Although the research question is well defined, the methods are not (see general comments). In general the bioinformatics and statistics part is poorly outlined, perhaps indicating the un-complete understanding of the performed analyses.

Validity of the findings

Based on poory outlined method section, I have no comments here.

Additional comments

L112 – it weird to say specimen for a stool. Perhaps simply “one stool sample”.
L 115 – each sample (outer surface, central inner and mixed stool) was 300 g??? A child’s stool can’t be ca 900 grams.
L118 – RNAlater and EtOH samples also fozen? Why???
L123 – “an attempt was made to keep samples in cool …” – was this just an attempt or did it really happen? Not clear.
L126 – how long was the freeze-thaw cycle?
L138 – “weighing process” – please specify here how many grams were included to the DNA extraction?
L147 – odd primers. provide primer names! Did not find those primers in Alcon-Giner et al. 2017.
L169 – provide indices
L171 – 8 cycles seems a bit low. Why only 8 cycles?
L182 – fix to 2 x 300 bp
L183 – “up to 100,000 reads per sample” specify reagent kit instead.
L184 – no need to specify the Python download link
L184 – cutadapt uses Python3, not 2.7! And I guess you ran cutadapt through some Linux system, not through Python itself?
L185 – “reads trimmed to a minimum quality score of 20” - not saying that you should analyze the data again, but in my opinion this is really bad quality trimming habit. Just average and only q>20 means that probably your reads still contain very high proportion of errors
E.g. the average q score of (1,4,7,9,10,20,20,20,30,30,30,30,30,40,40,40,40,40,40,40,40,30,30,30,30,10,10,9,5,5,5,4,3,2,2,2) is 20.5! But obviously the read has lot of errors.
L187 – “PANDAseq in Python version 2.7” – it’s very misleading if you say that pandaseq works like this on python!
L190 – “Operational Taxonomic Units were assigned with 97% clustering” – which algorithm? do not see that in CodeS1 file.
L195 – how was the UniFrac distance calculated?
L204 – “more significant p value” -> higher p value
L218 – “Backward elimination was used for sequential removal of non-significant variables (p ≥ 0.05)” – this is not how the backward elimination works, by just p value. Was the AIC model selection used with the backward elimination strategy?

Methods: Why did authors choose 2-step PCR and in between also purifying the samples? Very laborious. Any specific reasons for that?
Any measures taken to account for unequal sequencing depth between samples? Any transformations for the raw read count prior making similarity matrix?

L238 – report results of the statistics here as well.
L242-244 – how can an ordination method say statistical significance?
L289 – if PCA does not show any “apparent” clustering, it does not mean immediately that there is no difference between ‘groups’! This has to be tested in order to say something about “significance”.
L304 – “(p = 2.871e-13)” – p < 0.001 is enough.


CodeS1 file:
poorly organized script file.
“-q minimum quality (20 for pilot will increase to 25 for main study)” – what is a pilot, what is main? Carefully edit your files before submitting.
CodeS2: please edit for better vizualisation.

Supplementary tables should have legend inside the file.

---

## Round 0.2 · accepted · Accept

Dear Lauren and co-authors,

I am delighted to accept your revised manuscript for publication in PeerJ. Two of the reviewers have confirmed that you have address all previous concerns to their satisfaction, and I have also reviewed the manuscript myself - please find attached some minor comments that may be incorporated in the proofs.

Congratulations and thank you for this nice contribution!

With warm regards,
Xavier

Reviewer 1 ·

Basic reporting

well written

Experimental design

Now fine

Validity of the findings

Important technical development that should have wide appeal to the microbiome community

Additional comments

The authors have addressed the issues raised

·

Basic reporting

no comment

Experimental design

no comment

Validity of the findings

no comment

Additional comments

I have read the revised manuscript, and the authors have considered the comment and suggestions that I raised. The revisions that they made to the manuscript are very effective in addressing the remaining concerns. Unless noted otherwise from editors or co-reviewer, the manuscript could be accepted for publication.